# Experimental evidence of climate change extinction risk in Neotropical montane epiphytes

**Emily C. Hollenbeck** [1,2] ✉ **& Dov F. Sax**[1]

Climate change is conjectured to endanger tropical species, particularly in biodiverse montane regions, but accurate estimates of extinction risk are limited by a lack of empirical data demonstrating tropical species' sensitivity to climate. To fill this gap, studies could match high-quality distribution data with multi-year transplant experiments. Here, we conduct field surveys of epiphyte distributions on three mountains in Central America and perform reciprocal transplant experiments on one mountain across sites that varied in elevation, temperature and aridity. We find that most species are unable to survive outside of their narrow elevational distributions. Additionally, our findings suggest starkly different outcomes from temperature conditions expected by 2100 under different climate change scenarios. Under temperatures associated with low-emission scenarios, most tropical montane epiphyte species will survive, but under emission scenarios that are moderately high, 5-36% of our study species may go extinct and 10-55% of populations may be lost. Using a test of tropical species' climate tolerances from a large field experiment, paired with detailed species distribution data across multiple mountains, our work strengthens earlier conjecture about risks of wide-spread extinctions from climate change in tropical montane ecosystems.

Tropical mountains are the most biodiverse regions on earth[1], and tropical montane species are conjectured to be at high risk of extinction from climate change[2-5]. These risks may be particularly acute in tropical montane regions, where species have narrow ranges[6], and the climate is changing more quickly than in the lowland tropics[7]. To avoid extinction, many tropical species may need to shift their distributions upward along elevational gradients[3,5,8-10]. Such upward shifts in distribution are a commonly expected response to climate change[11], and aggregate vegetation types have begun to shift uphill in some habitats[10,12]. However, individual species appear to be migrating less rapidly than expected to keep pace with climate[13], particularly in tropical regions[14,15]. Overall, the concern over tropical extinction risk is amplified by a deficiency of data on species' distributions and climatic tolerances from these diverse and complex ecosystems.

Species Distribution Models (SDMs) are an important tool to assess whether species can survive projected changes in climate within their current distributions, or if they will need to migrate to track their climatic niche[16-18]. Forecasts based on SDMs can be useful, but their accuracy in tropical montane regions is compromised by data limitations[19]. For example, they depend on the assumption that species' distributions are in equilibrium with climate, but few experimental studies have tested tropical species' climatic tolerances. SDMs also require ample occurrence data to be accurately matched with climate metrics. However, most species' distributions are poorly documented in the tropics, and the abrupt environmental heterogeneity of these mountains is poorly matched to the spatial scale of climate data that are typically available, limiting the effectiveness of SDM-based approaches in the world's biodiversity hotspots[19-21]. These constraints in tropical montane systems have led to longstanding calls

[1]Department of Ecology, Evolution, and Organismal Biology & Institute at Brown for Environment and Society, Brown University, Providence, RI, USA. [2]Tiger Works Research & Development, Avenues: The World School, New York, NY, USA. ✉e-mail: Hollenbeck.ec@gmail.com

for more experimental work to empirically document species tolerance to climate conditions[9] and for more fine-scale survey work to better document species distributions along elevational gradients[8].

Both calls for additional work have focused attention on epiphytic plants, which can represent up to 50% of the vascular plant diversity in tropical montane habitats[22]. Epiphytes, which grow perched on tree branches and trunks rather than rooting in the ground, are believed to be particularly vulnerable to climate change because of their growth form, narrow elevational ranges, and sensitivity to drought[3,23–26]. Recent work suggests many epiphyte species are at risk of extinction from climate change[27]. Some of the most compelling applications of SDM-based forecasts of extinction risk have built upon fine-scale surveys of epiphyte distributions in tropical montane systems[28,29]. Epiphytes have also been the subject of transplantation studies that have aimed to experimentally validate the climatic tolerance of individual species. Existing experiments have focused on a relatively small number of species, have been of short duration, and haven't generally featured a reciprocal transplant design. Nonetheless, they have reinforced the contention that epiphyte species have narrow climatic tolerances, with a particular sensitivity to drought[30–34]. We address these research gaps by pairing detailed surveys of species distributions along elevational gradients together with experimental transplant experiments that directly tested species' climatic tolerances. The experiments involve reciprocal transplants, across multiple sites and climatic gradients, and involve sufficient numbers of species and replicates to draw general conclusions. These experimental results are leveraged with surveys of species' distributions across multiple elevational gradients, in order to contextualize extinction risk at both local and regional scales.

Here we describe estimates of species extinction risk from climate change based on our findings from a long-term, reciprocal transplant experiment of vascular epiphytes at Monteverde, Costa Rica, combined with fine-scale surveys of species distribution across three montane areas in Costa Rica and Panama. Monteverde is an iconic location for studying mesic, tropical montane cloud forest ecosystems, and is the site of one of the first documented global extinctions for which climate change was a hypothesized cause[35]. The mountain's complex topographical interactions with cloud cover enabled us to examine two distinct climatic gradients, one in which aridity varies but temperature is relatively constant, and another in which aridity and temperature covary (Fig. 1a and Fig. S1). Along these two gradients, we established six reciprocal transplant sites (Fig. S2). We transplanted 15 different species, comprising more than 1500 individual specimens (Table S1), which we monitored over the course of three years for survival, herbivory, and abiotic stress. We selected study species from two genera, *Elaphoglossum* and *Peperomia*, that are widespread, taxonomically diverse in the Neotropics, and could be readily identified in the field[36]. We conducted fine-scale surveys of the elevational distribution of 67 of these species across four elevational gradients: the Pacific and Atlantic slopes in Monteverde, the Barva transect, and Volcan Baru (Fig. 1b, c and Fig. S3). These data allow us to contrast extinction risk estimates on individual mountains and in aggregate across the region, using multiple dispersal scenarios and alternative projections of climate change. We demonstrate high vulnerability to climate change in tropical montane epiphytes.

## Results
### Transplant experiment
Our reciprocal transplant experiments showed that epiphyte species struggled to survive outside their native ranges. This was true when species were moved from cool, wet conditions to warmer or dryer conditions (e.g., *Peperomia palmana*, Fig. 2a), and when species were moved from warm, dry conditions to cooler and wetter conditions (e.g., *P. dendrophila B*, Fig. 2b). Patterns varied among species and transplantation sites, but 12 of 15 species experienced significantly higher mortality away from native sites. Of the remaining three species, the increase in mortality outside the native range was nearly significant for one ($p = 0.08$) and non-significant, but still apparent, for two species moved from the warmest site to cooler, wetter sites (Fig. S4 and Table S2). These patterns hold when species are viewed in aggregate (Fig. 2c) and suggest high, but asymmetrical sensitivities to changes in temperature and aridity: Drier and warmer changes had a greater impact than cooler and wetter. When considering relative changes in climate using intervals of change (see Methods, Fig. S1), all species transplanted to cooler and/or wetter sites one interval away from source populations were able to survive, whereas most species could not survive a move of similar magnitude towards warmer and/or drier sites (Fig. 2 and Table S2). Most species survived when moved to slightly drier sites if temperature was unchanged, but any combination of increased aridity and temperature resulted in increased mortality

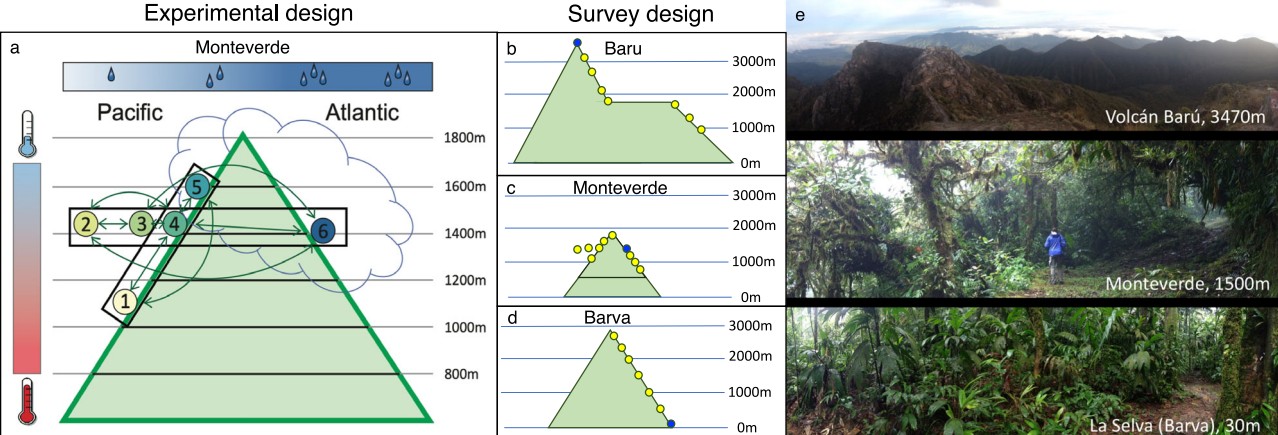

**Fig. 1 | Experimental and survey design. a** In Monteverde, reciprocal transplants were performed with 15 epiphyte species across six sites along two intersecting gradients: one that varied in elevation, temperature, and moisture, and another that varied in aridity, but held elevation and temperature relatively constant. **b–d** Four elevational transects were surveyed across three mountains in Costa Rica and Panama, to determine the elevational ranges of 67 species with thorough presence-absence data recorded from individual study sites (circles). **e** Individual study sites had ecosystem characteristics that varied along the elevational gradients, ranging from high-elevation sites above tree line (e.g., Volcán Barú), to mid-elevation cloud forest (e.g., Monteverde), to low-elevation tropical rainforest (e.g., at the base of the Barva transect in La Selva). Photographs in (**e**) (all taken by the authors) illustrate specific sites indicated with blue circles in (**b–d**). In all panels, "m" labels indicate meters above sea level.

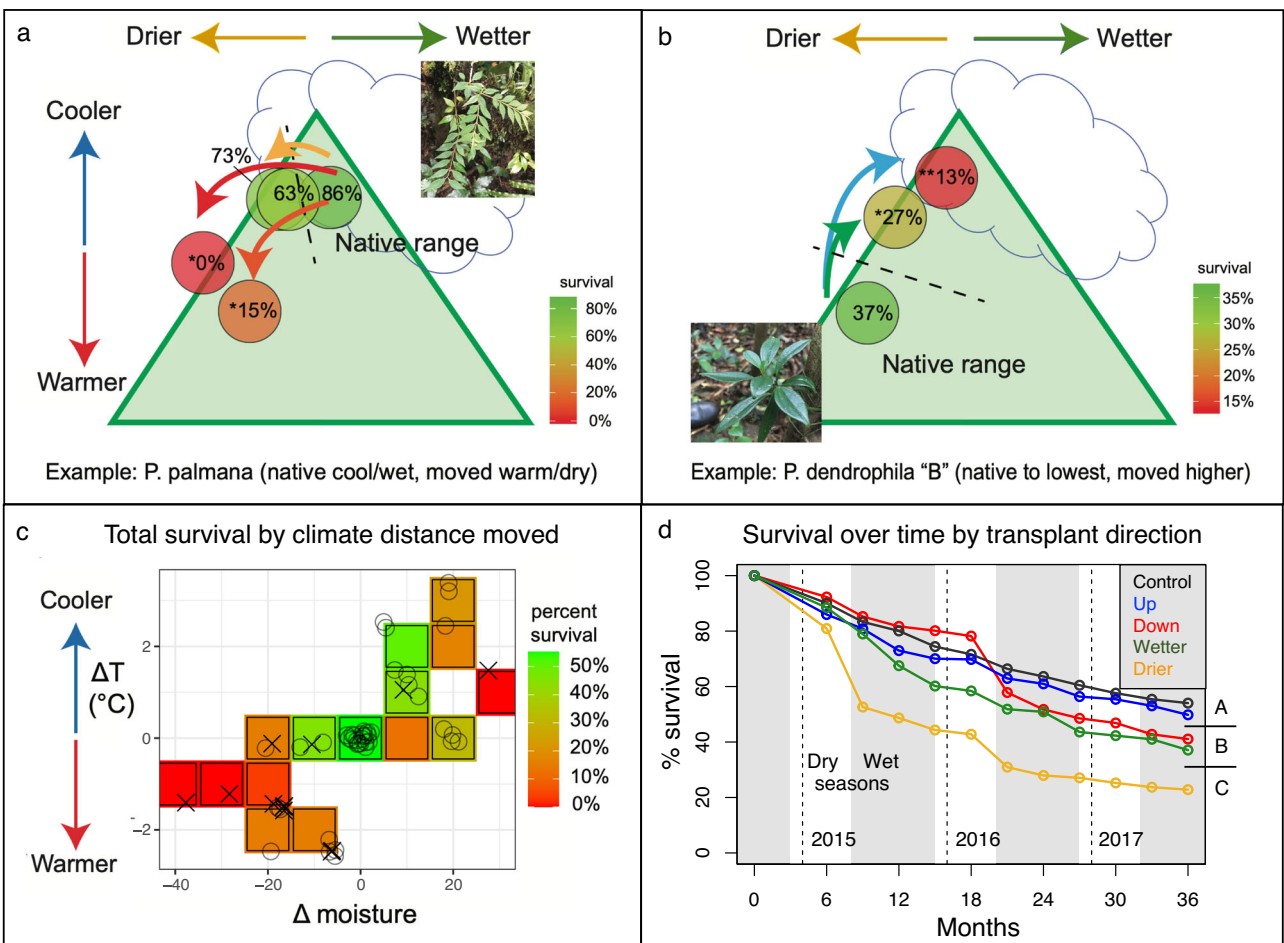

**Fig. 2 | Transplant experiment shows high mortality for epiphyte species outside their native range.** Top panels show transplant survival after 3 years for two example species, one moved downhill to warmer and drier sites than its native range (**a**), and the other moved uphill to cooler, wetter sites (**b**). Circles represent transplant destination sites. Photos taken by the authors. Percent survival is shown at each destination site, with asterisks indicating significant differences from the native site(s), and color indicating high (green) to low (red) survival. **c** Survival across all 15 species after 3 years is synthesized with a "heat map," showing each species' survival based on the quantitative distance from their native range boundary across gradients of temperature (°C) and moisture (leaf wetness; see Methods). Survival at all native sites is averaged at (0,0). Locations of each unique species movement by climate-distances are shown either as a circle (indicating that some individuals of a species survived) or as an "x" (no individuals of a species survived), with scatter to aid visualization; all values within each grid cell are averaged (colored squares) to show overall trends. **d** Points show the average survival of all transplants, measured at 3-month timepoints, based on the qualitative direction they were moved across the climatic gradients. This includes plants transplanted higher (Up) or lower (Down) along the elevational transect; transplanted to sites with climates wetter (Wet) or drier (Dry) along the moisture transect; and plants that were manipulated but ultimately deposited at their native site (Control). Vertical gray bands illustrate the approximate boundaries of the wet season (May-December); vertical dashed lines indicate January 1 of each year. Statistically significant differences between groups, calculated using Kaplan–Meier curves (see Methods), are indicated by the letters A, B, and C. Transplants moved to sites drier or lower than their sites of origin show increased rates of mortality during the dry seasons of the first two years of the experiment. Source data are provided as a Source Data file.

(Fig. S4). Among the few species transplanted more than two intervals from their source population, none survived (Fig. 2c). Background rates of mortality were high in both unmanipulated and experimental controls (Figs. S4 and 2d), which is consistent with high mortality rates of epiphytes noted in other studies[37]. However, mortality was significantly higher in transplants moved to different climates relative to experimental controls, demonstrating the relative importance of changes in aridity over other factors in driving transplant mortality (Fig. 2d).

While most of our study species have narrow elevational distributions, there were three widely distributed species that were each reciprocally transplanted in 13 or more pairwise donor-recipient sites. Each of these species showed increased mortality at drier sites within their native range. One of these species, *P. peltilimba*, showed an effect of population of origin, likely due to local adaptation or parental effects, such that individuals from the wettest site had significantly

reduced survival when transplanted to the driest site within its native range (Fig. S6 and Table S3). Two species had significantly reduced survival at both wet and dry peripheries of their own native ranges, with no effect on the population of origin (Fig. S7 and Table S4). This result suggests that these peripheral sites (both wettest and driest) have sub-optimal climate conditions for these species.

Reduced survival in our experiment at drier sites appears to be caused by abiotic stress, and reduced survival at wetter sites by an interaction between abiotic and biotic stress. Competition was unlikely to impact mortality due to experimental design, as transplanted species were placed on wooden scaffolds with ample space between individuals (Fig. S2). Herbivory pressure is relatively low at these sites (Fig. S8), and transplantation did not cause consistent changes in herbivory (Fig. S9 and Table S5). Indeed, the highest baseline herbivory occurred at the site of the highest overall survival, while the lowest herbivory rates occurred at the driest sites with the lowest survival (Fig.

S9b). These results suggest that herbivory, while present, is relatively unimportant in driving species distributions—a result consistent with other studies, which generally find little impact of herbivory on tropical epiphytes[26]. Mortality at warmer and drier sites was often preceded by visible desiccation. In contrast, mortality at wetter sites was often associated with rotting tissue, a moisture-related biotic pressure that could potentially be accentuated by stress from reduced gas-exchange ability in wet leaves. Finally, the importance of climate in driving mortality is also supported by the timing of mortality events, which occurred disproportionately during the season of greatest climatic stress at a given site. Transplants tended to die during the dry season at the dry sites and during the wet season at the wet sites (Fig. 2d).

Monteverde is an ideal site for these experiments because it contains sharp microclimatic variation over steep gradients, allowing observation of the impact of climates over relatively short distances and differences in elevation (Fig. 1a and Fig. S1). No individuals of any species survived if moved across the maximum elevational difference among transplant sites, which is just 500 m elevation. Likewise, no individuals of any species survived if moved from the wettest to the driest sites along our aridity gradient, in which elevation and temperature were held relatively constant: all four sites on this gradient fall within 100 m elevation and 4 km distance from each other. These dramatic results on such a fine scale highlight the sensitivity of these species to changes in climate. They also speak to the pronounced climatic differences that can occur in tropical montane systems across relatively short geographic distances. For example, our wettest and driest sites showed order of magnitude differences in the number of days in which conditions for epiphyte leaves were fully wet or fully dry (Fig. S1). These specific types of differences in aridity are important for epiphytes at Monteverde, which, during the dry season can maintain their water balance through foliar water uptake[38,39]. Indeed, differences in cloud cover are known to be particularly critical in determining where montane-species occur and are one of the most important measures of aridity in these ecosystems[2]. Ultimately, our results are important relative to the extinction risks tropical epiphytes face from the specific changes in climate forecast to occur in montane regions of Central America, where both temperature and aridity are projected to increase[40–42].

## Elevational range surveys

A key component to estimating extinction risk for species in these systems is accurately documenting their elevational distributions and current range limits. Our fine-scale surveys for 67 species in our two study genera showed significant variation in the distribution and abundance patterns for individual species on the three mountains surveyed (Tables S6–8). While some species occupied large elevational extents, most species had relatively narrow elevational distributions on each mountain where they occurred, as well as in aggregate across the surveyed mountains (Fig. 3). The average distributions of species at each of the individual mountains surveyed were 489 m at Monteverde, 704 m at Volcan Baru, and 858 m at Barva. Species' mean elevational distributions, when aggregated across all three montane areas were 1076 m.

To estimate the impact of climate change on montane species, we consider how far uphill species will need to shift to maintain their current conditions. This common approach[3,8] is relevant here as the cool, wet conditions that support most epiphyte species[24] are expected to shift upwards in elevation[40–43]. Our transplant experiment affirmed that these epiphyte species are climate-limited at their lower-elevational boundary, an important result that enables us to uphold this assumption for the following analysis. We calculated how these species would be impacted by a 600 m shift in elevation, which corresponds with the 3.2 °C warming expected in the region by 2080 under SSP3-7.0[44] (Fig. 3). With climate-limited ranges, populations whose lower boundaries currently occur within 600 m of the mountaintop would face mountaintop extinction, meaning the populations would be extirpated from that mountain[5,8]. This analysis showed that a 600 m shift in elevation would result in the extirpation of 12% of species at Monteverde and 18% at Barva, but no species loss at Volcan Baru, which extends to higher elevations (Fig. 3). Additionally, species with elevational distributions of less than 600 m breadth would be at risk of extinction if they cannot shift their elevational distributions. These species are defined as having range-shift gaps because there is an elevational gap between their current and projected elevational-ranges[3]. Such gaps are projected to occur when no part of the species' current distributions remains climatically suitable, forcing these species to migrate to entirely new elevational ranges in order to remain within their climate niches[3,8]. If these narrow-ranging species are both unable to tolerate the warming climate and also unable to shift their distributions, then they would be extirpated from individual mountains as well (Fig. 3). It is important to note that these two categories, those with lower range boundaries within 600 m of a mountaintop and those with narrow ranges of less than 600 m, combine to include roughly 50% of all species on each of the three mountains surveyed (Fig. 3). When species' aggregate elevational distributions are considered across all three mountains, 23% of species fall into one of these two risk categories (Fig. 3). Many of the remaining species, not in either of these categories, are nevertheless projected to lose half or more of their existing elevational range, as their ranges contract at mountaintops, putting these species at increased future risk of extinction (Fig. 3 and Fig. S10).

To provide a more thorough assessment of extinction and extirpation risk, we bracket our estimates using alternative scenarios: the amount of warming, dispersal capacity, and local vs. aggregate range sizes. All three of these distinctions significantly impact the projected rates of population extirpation and species extinction, allowing us to describe the impact of these factors on extinction risk (Fig. 4). We present forecasts along a continuum of warming (Fig. 4a, b), and call out impacts at 1.5 and 3.2 °C in order to illustrate specific scenarios of low vs. mid-high emissions. We demonstrate the risk difference if species have full dispersal capacity, meaning only mountaintop extinction species will disappear, vs. no dispersal capacity, meaning all species with range-shift gaps will also disappear. Finally, we test two ways of quantifying each species' elevational range: local ranges indicate a population is considered restricted to the elevations where it was found on each mountain, while aggregate ranges consider the full breadth of elevations a species occupies across all mountains (see Methods). Extinction in this context means a species will be gone from all three surveyed mountains, while extirpation means a population is eliminated from a single mountain.

The difference in risk among the full factorial set of scenarios is dramatic. For example, at 1.5 °C of warming, full dispersal, and aggregate ranges, no extinctions are predicted. Whereas 36% of species are expected to go extinct with 3.2 °C of warming, no dispersal, and local ranges (Fig. 4d). Qualitatively similar differences among alternative combinations of scenarios are observed for population extirpations. The largest differences in outcomes are driven by warming and dispersal capacity, with a smaller impact of whether local or aggregate elevational distributions are considered. Under the most severe combination of scenarios, more than 50% of the populations of these epiphyte species could be lost (Fig. 4c).

## Discussion

The true number of populations and species extirpated from these mountains will depend upon a variety of factors that collectively suggest the higher end of our bracketed estimates is more likely. First, the particulars of future climate change will be more complex than a simple rise in temperature, and may exacerbate risk to epiphytes. Reductions in cloud moisture may be greater than that expected from simple elevational shifts in climate[43], which would increase aridity and

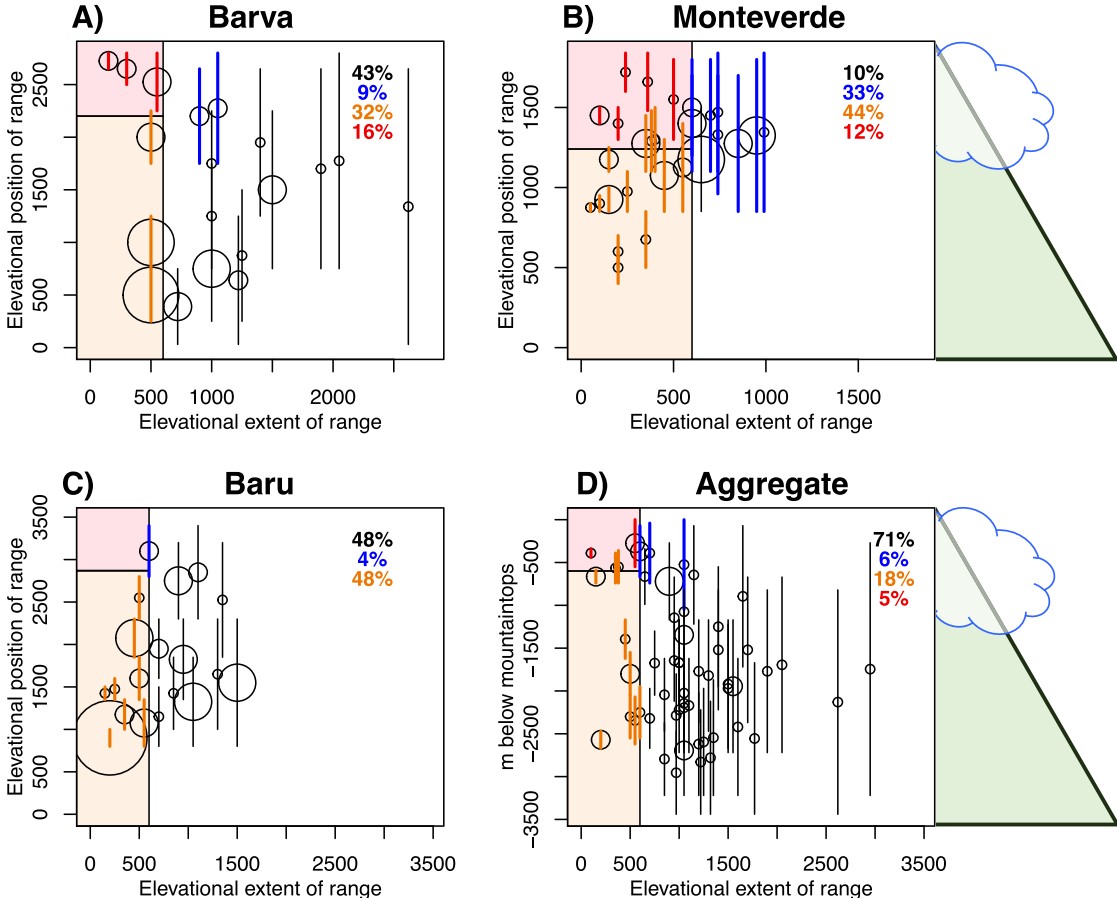

**Fig. 3 | Species' elevational ranges are used to estimate extinction risk with 3.2 °C warming.** A–C illustrate species' ranges in meters above sea level on individual mountains, while **D** considers each species' elevational range as the sum of its ranges on all mountains where it was found. In panel **D**, the position on the y-axis measures the distance (in meters) below the top of the highest mountain where each species was found, rather than total elevation a.s.l., to better illustrate the risk of mountaintop extinction as the three mountains have different peak heights. Species are illustrated with a circle, with the circle size proportional to the number of species it represents (ranging from 1–8). Species cluster together because of discrete sampling sites. Vertical lines show the breadth of the native elevational range for each species or species cluster, with the circles placed at the midpoint of these distributions on the y-axis. The y-axis represents the distance below the highest local mountaintop for each species. Red species have elevational ranges restricted to within 600 m of the mountaintop (shaded red box on the top left of each panel) and are projected to lose all their available climate space, becoming extinct. Orange species have elevational ranges narrower in breadth than 600 m, and will face extinction if they cannot disperse between their current and future range. Blue species will lose half or more of their current elevational range, because their range midpoints fall within 600 m of the mountaintop. Source data are provided as a Source Data file.

thus extinction risk in these moisture-sensitive species. Warming could be greater than predicted, because temperatures are rising more quickly at high elevations[6,45], or due to human failure to mitigate greenhouse gas emissions. Dispersal may also be a challenge in this system, which would push our risk estimates towards the higher end. Available evidence suggests that tropical, montane species have low dispersal ability[6]. Dispersal limitation may be particularly acute above tree line, as tropical high-elevation trees are showing lags in their ability to recruit above the tree line in response to warming, which would provide an obvious barrier to epiphytes that depend on mature trees for their habitat[46]. Finally, our aggregate range size approach makes some assumptions that might downplay the true level of risk. For example, using aggregate ranges assumes that climate conditions are equivalent at the same elevations on different mountains, and that there is no local adaptation of populations on individual mountains. The less valid these assumptions, the closer true risks will correspond with estimates of loss based on local ranges.

Some speculative considerations could conceivably reduce extinction risk in this system. Evolutionary rescue may help some species[11,47], although the slow growth rates of our study species make it seem unlikely that they could evolve rapidly enough to escape climate-related extinction. Another possible mitigator would be the presence of micro-refugia that maintain cooler and wetter conditions where species could persist. Unfortunately, work to understand and predict where micro-refugia might exist are still at an early developmental stage[48] and it is difficult to evaluate whether they could play a significant role in tropical montane systems.

Finally, we estimated extinction risk based on our experimental results: assuming that species will not survive in the long-term in conditions much warmer and drier than their current low-elevation range boundary. Our estimates would inflate actual risk if our main conclusions from the transplant experiment were flawed. Importantly, there are limitations to extrapolating any experimental results, including our field experiments. For example, the 15 species examined might not be broadly representative of tropical montane epiphytes; experimental mortality could have been caused by factors in the field other than climate that were not detected; conditions for epiphytes in the understory might not follow the same pattern as in the canopy or for other organismal growth forms. Any of these considerations could weaken the generality of our broader conclusions, particularly if they lead to the erroneous attribution of climate to the mortality patterns observed across sites. Though possible, we believe this is unlikely. The

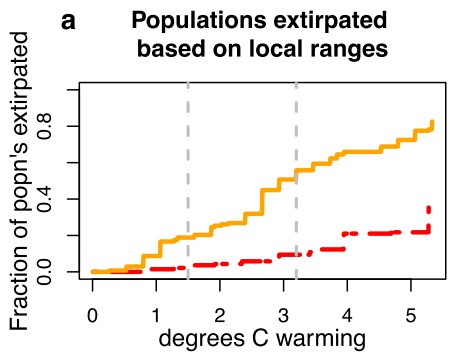

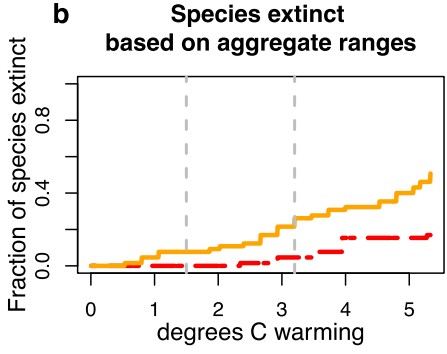

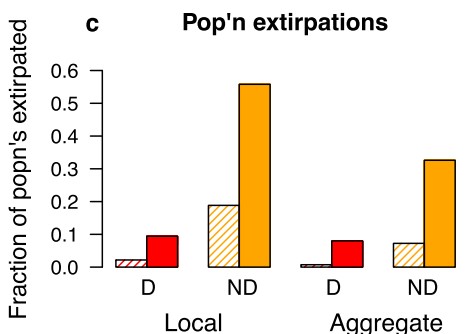

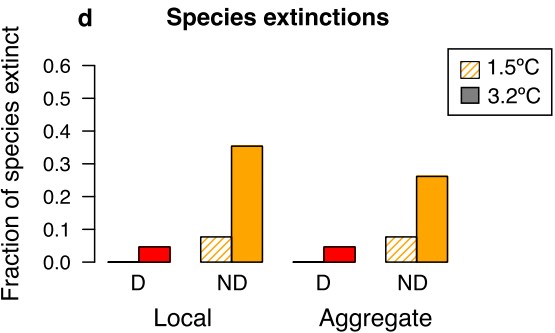

**Fig. 4 | The predicted fraction of population extirpations and species extinctions varies strongly among alternative scenarios.** Elevational distributions calculated for "local ranges" are based on species' individual distributions on a single mountain, whereas distributions calculated for "aggregate ranges" are based on the sum of ranges observed across the mountains where species occur (see Methods for additional details). Top row: lines show predictions for population extirpations based on local ranges (**a**) and species extinctions based on aggregate ranges (**b**) as a function of degrees of warming and either no dispersal (solid orange line) or full dispersal (dashed red line). Vertical dashed lines indicate benchmarks of 1.5 and 3.2 °C of warming. Note that warming beyond those benchmarks leads to increased estimates of population and extinction loss. Bottom row: histograms show a factorial suite of alternative estimates of population extirpations (**c**) and species extinctions (**d**) based on alternate scenarios of Dispersal ("D") vs. No Dispersal ("ND") and Local vs. Aggregate range breadth, and warming of 1.5 °C (hatched bars) vs. 3.2 °C (solid bars). Source data are provided as a Source Data file.

pitfalls that threaten experimental conclusions are reduced by our use of multiple in-range controls, adequate replication, many species, and results over multiple years[49,50]. While previous transplantation experiments on epiphytes[30–34] did not meet all these criteria, which are difficult to achieve in field conditions, their conclusions are broadly consistent with our own. We believe the coherence of our findings with those from related studies ameliorates a major limitation of drawing firm conclusions from any single study, including ours. Still, these conclusions could be strengthened in the future by additional transplant experiments including different types of species, or in different geographical areas, by examination of species' dispersal capacity, and by greenhouse studies that can carefully isolate individual climatic factors. We believe that our study provides a thorough transplant experiment from a tropical montane ecosystem, helping to fill an important gap in a data-deficient system.

Ultimately, it seems likely that the actual rate of extinction risk falls along the higher end of our bracketed estimates, since there appear to be more uncertainties that would increase rather than decrease risk. Our estimates are concerning, particularly at higher levels of climate warming, in the absence of dispersal, and considering local ranges, with more than 50% of epiphyte populations extirpated from individual mountains and more than 30% of species extinct across the surveyed region. These projections of extinction risk from climate change are just slightly higher than other recent estimates in tropical plants: epiphytes in Honduras[29] and Brazil's Atlantic Forest[27], and tropical trees in the Andes[18]. Our higher estimates of risk relative to other taxa are not unexpected, given the expectations that epiphytes might be particularly vulnerable to climate change[3,23–26]. Overall, our results are consistent with growing alarm about extinction risks in

many tropical montane taxa, including amphibians, birds, insects, and other taxonomic groups[5,51–55].

A common thread throughout all of this work is concern that narrow climatic tolerances of these tropical montane species will put these species at particularly high risk of extinction[3,5,29,53,56]. We believe that our thorough, multi-year transplant experiment provides the empirical evidence to substantiate this long-held contention, and that our survey work across three mountains broadly contextualizes this risk. While we cannot be sure that our study taxa are fully representative of other epiphytic taxa, or more generally of other vascular plants, the strength of our results adds to growing concern that pronounced climate change will lead to massive biodiversity loss in tropical montane systems and emphasizes the critical role that climate change mitigation can play in saving species from extinction.

## Methods

### Ethics and regulations

All of our field research was approved and permits granted by the relevant government institutions in the two countries where fieldwork was performed: Costa Rica's Sistema Nacional de Áreas de Conservación from 2013 to 17 (Scientific Passport no. 04964), and Panama's Ministerio de Ambiente in 2016, as well as the landowners of all private reserves and properties where data were collected (see Acknowledgements), as required by the government permits.

### Study species

This study focuses on species in the primarily epiphytic genera *Elaphoglossum* (ferns, fam. Dryopteridaceae) and *Peperomia* (dicots, fam. Piperaceae), which have high diversity in tropical montane forests

(around 700 and 1700 species described worldwide, respectively). We collected data on all plants found within these genera during field-work, but restricted analyses to 67 species that could reliably be identified in the field for detailed distribution surveys. Sixty-four of these species were identified at the species level and three of them, each with highly restricted distributions, were identified as morpho-logical species (Tables S6–8). We selected 15 species (see criteria below) for study in reciprocal transplant experiments. Voucher specimens of all study species were deposited in one or more herbaria: the National Herbarium of Costa Rica, the Herbarium of the Smithsonian Tropical Research Institute in Panama, and the Herbarium of the University of Panama. Voucher specimens from this study can be identi-fied as all specimens in the above herbaria collected and deposited by Emily Hollenbeck from the years 2013–2017.

## Study area context

Species distribution data were collected by the authors across eleva-tional and climatic gradients on three mountains in Costa Rica and Panama: Monteverde, Volcan Barva, and Volcan Baru (Fig. 1 and Fig. S3). Monteverde is one of the best-studied cloud forests in the world, where the impacts of climate change have been demonstrated for decades. Documented abiotic changes have included an increase in precipitation variability, longer and more frequent drought periods, and higher nighttime temperatures[57]; all these trends have continued since they were first reported in 1999. Biological changes include upward range shifts of birds and reptiles, amphibian extinctions and extirpations, and reptile and amphibian population declines[57].

## Climate data and profiles of field sites

See Fig. S1. We collected climate data at all sites in Monteverde for time periods ranging from 6 months to 3 years. LogTag data loggers at all sites recorded hourly temperature and relative humidity (RH), while Decagon data loggers at transplant sites recorded temperature, RH, precipitation, and leaf wetness (LWS). Precipitation data were not usable because precipitation gauges were frequently clogged with debris. LWS is measured by a plastic leaf-shaped sensor that records the number of minutes per hour that the sensor surface is wet. Sensors were secured at approximately a 30-degree angle from the horizontal to mimic a natural leaf position. "Dry days" occurred when the LWS sensor was dry for the entire day (i.e., recorded 0 min wet for every hour in that day), while "wet days" occurred when the LWS sensor was wet for the entire day (recorded 60 min wet for every hour in that day). The number of dry days and wet days were calculated as the total number of days from over 11 months of continuous LWS data (6/12/2015-5/24/2016).

Sensors frequently fail in these extremely humid environments, so there are gaps in the climate data. Comparisons between sites are calculated from time periods with continuous data from all sites. Lapse rates provided are calculated from a 6-month period (September 2014–March 2015) when all ten sites had simultaneously functioning LogTag sensors. Temperature and moisture profiles for transplant sites are calculated from an 11-month period (June 2015–May 2016) when both temperature and LWS sensors were working for all six sites (Fig. S1).

We report an adiabatic lapse rate of 5.33 °C per 1000 m elevation in Monteverde and use this for calculations in this study. Lapse rates reported from field climate measurements on Volcan Barva range from 5.1 (2) to 5.5 °C per 1000 m[58].

## Transplant experiment

**Species selection.** Fifteen species of *Elaphoglossum* and *Peperomia* are reported in the transplant experiment (Fig. S4). Two additional species were used in the experiment, but had such low sample size that no statistical power could be deduced, so they are excluded from analysis. From the species that were detected at Monteverde during

the surveys (see below), we selected species for the transplant experiments if they met the following criteria: could be found growing naturally in the forest understory (not canopy specialists), small enough to be transplanted without serious disturbance (see Methods below), and abundant enough for several dozen individuals to be collected from a single site while leaving sufficient individuals at the site to be monitored as unmanipulated forest controls.

**Transplant experimental design and methods.** All transplant experiments were conducted at Monteverde. Reciprocal transplants were performed across two distinct climatic gradients: an aridity-only gradient, encompassing four sites at similar elevations (1380–1480 m), but very different moisture regimes; and a Pacific slope elevational gradient, with three sites from 1100–1600 m encompassing gradients in both temperature and precipitation (Fig. 1a and Fig. S1). Across each transect, transplants were performed reciprocally between all sites along that transect. We define a "direction" of transplant movement as movement from a unique origin site to a unique destination site (e.g., each arrow in Fig. S1).

Wooden scaffolds were constructed to be the substrate on which the epiphytes would be moved and persist through the duration of the experiment (Fig. S2). Scaffolds were constructed from 1 × 2" planks of untreated laurel wood. Scaffold architecture allowed for epiphytes to be placed on either horizontal or vertical beams, or at beam inter-sections, replicating the variety of natural orientations in which they were found growing. Only epiphytes growing in the forest understory (on low trunks or branches, or surviving and thriving on fallen bran-ches) were collected. Transplanted epiphytes were tied to the scaffold with small strips of nylon fabric, a method commonly used in orchid cultivation. When collected epiphytes were growing rooted in a sub-strate of canopy soil, a piece of that substrate surrounding the plant's rhizome or root system was cut out and transplanted along with the plant.

Each unique transplant direction (from one site to another) was realized with four replicate scaffolds, which were approximately identical in species composition and number of individuals per spe-cies, but not the physical layout of individuals/species on the scaffold; e.g., epiphytes were placed randomly on each scaffold while still accounting for individual architecture (whether a plant was growing on a horizontal or vertical branch), and individuals were spaced spread apart relatively evenly across scaffold beams so that they would not be directly competing for light or space. The ideal sample size was roughly 20 individuals per species per direction, with five individuals on each replicate scaffold. In many cases, not enough individuals could be located in nearby forests, so a lower sample size was used, with an average of 13.6 individuals used per species per direction, with a range from 4 to 22 individuals; exact sample sizes per species per transplant direction are given in Table S1.

Each site was divided into four replicate "microsites" placed along a 100 m transect in a flat area of the site. This was done to account for the possibility of subtle microclimatic variation, as well as to offset the risk of losing some scaffolds due to tree and branch falls or other natural causes. Microsite location was selected using a random num-ber generator from 0 to 100, and placed at that distance along the transect. The four replicate scaffolds coming from each unique origin site were distributed among the four microsites. Thus, at each unique destination site, each microsite had identical scaffolds in terms of scaffold origin and composition. Among the 96 total scaffolds used in the experiment, three were lost before the end of the 36-month experiment due to being crushed by fallen branches. All three lost scaffolds were at site 5; two at the same microsite (from origin sites 1 and 5) after 30 months, and the third at a different microsite but also from origin site 5 (e.g., the experimental control), after 24 months. Interval censoring of survival data (see Analysis) allows the data from these scaffolds to remain incorporated into survival curves prior to

their destruction, and their removal from the experiment does not factor into calculated mortality estimates.

Transplants were initiated during the rainy season in July–August 2014, when epiphytes were collected and tied onto scaffolds. All scaffolds were left to acclimate for 3–8 weeks between plant setup and movement, and scaffolds were moved to destination sites from September 1–8, 2014. Scaffolds were moved by being carried on the shoulders of two people, who hiked through the forest along trails between sites. Sites 1, 2, and 3 had to be accessed by road as well as trails, so all scaffolds moved to or from those sites were transported by car (for 5–20 min), with great care taken to not touch, disturb, or bounce the plants. Sites 4, 5, and 6 were connected by continuous trails.

Two types of control treatments were utilized in this experiment. First, experimental controls were treated exactly as all other transplants, i.e., were tied onto scaffolds and carried through the forest for a comparable amount of time to other transplants (30–60 min), but were then deposited at the site of origin. Second, unmanipulated or "forest" controls were plants tagged and monitored in the forest understory without ever being touched. The average survival of forest controls was 40% across all species and sites, which was lower than experimental controls (52%), and forest control survival was idiosyncratic and varied widely by species (Fig. S5). We observed that forest control mortality tended to occur for different reasons than mortality in experimental plants, such as their substrate (branches or trunks) breaking or being disturbed, being crushed by fallen organic material, or simply disappearing from their substrate without a trace. In contrast, experimental plants tended to die in place and could be observed before and after mortality, often appearing to desiccate or rot slowly over the course of multiple observations. From monitoring these control plants, we learned that epiphyte mortality in the forest understory is naturally high due to disturbance events, and thus highly idiosyncratic. For this reason, and because forest control survival was lower than experimental control survival, we do not adjust our experimental survival rates according to control survival for each species, as this would reflect idiosyncratic disturbance events rather than climate-induced mortality. We nonetheless present the forest control data (Fig. S5) to illustrate that the survival of experimental controls was higher than that of forest controls, and, thus, the experimental transplant procedure itself is unlikely to have inflated mortality rates.

Data on the survival, reproduction, growth, and herbivory of the transplants were collected quarterly for three years, starting immediately after transplant (September 2014), for the duration of the experiment (until September 2017). For every plant at every time point, a photo was taken of the plant, and survival and the number of reproductive structures were recorded. *Peperomia* produce spike-like infloresences/infructesences with many microscopic flowers and fruits, so the number of infloresence spikes per plant were counted. *Elaphoglossum* produces dimorphic fertile leaves that are morphologically and developmentally distinct from the sterile leaves, so the number of fertile leaves per plant were counted. Growth and herbivory were recorded for a subset of all plants: one individual per species per scaffold was randomly selected and then repeatedly measured. In the event that the individual being monitored died, a new individual of that species was selected from that scaffold and monitored for as long as it was alive; this process was continued unless all individuals of a particular species on a particular scaffold had died. Growth was measured by counting all leaves on each plant. Herbivory was measured by counting all the leaves with visible herbivory damage and estimating the amount of leaf tissue lost on a scale of 1–5, such that 1 corresponded to between 0–20% of tissue lost (average 10%), 2 means 21–40% (average 30%) of tissue, etc. The overall herbivory score for that plant was then calculated as the total fraction of leaf tissue lost for that plant:

$$H = (\text{\# of leaves with herbivory})*(0.2*\text{herbivory score}-0.1)/(\text{total \# leaves})$$

## Transplant experiment

**Statistics and analysis.** All data analysis was carried out using R[59].

Transplant survival data: Although species-specific transplant survival is visualized using only the final survival values (after 36 months) for simplicity (Fig. 2 and Fig. S4), significant differences in survival between groups were determined using survival curve analysis. Survival was analyzed with non-parametric likelihood estimators (NPMLE), more commonly known as Kaplan–Meier (KM) curves, for interval-censored survival data using the R package "interval." For each transplant species, KM curves were constructed for survival at each destination site. The impact of the destination site on survival was assessed for each species using an asymptotic log-rank trend test[60].

Herbivory data: Herbivory results were analyzed using ANOVA after a log(data + 1) transformation of herbivory values (Figs. S8, S9 and Table S5). Two separate metrics of herbivory were assessed: baseline herbivory at the beginning of the experiment, and change in herbivory after transplant. These were both calculated for each individual plant with herbivory measurements. Baseline herbivory was the initial herbivory measurement at the start of the experiment. The post-transplant herbivory rate was averaged across all measurements from 1-3 years post-transplant. The change in herbivory was the post-transplant rate minus the baseline rate. For plants that we began monitoring during the course of the experiment (due to replacing a deceased individual), we assigned them a baseline herbivory value of the average of that species at their same origin site.

## Elevational range surveys

**Survey design and methods.** Species occurrence data were collected between 2013 and 2016 on the mountains of Monteverde and Volcán Barva in Costa Rica, and Volcán Barú in Panama (Fig. 1 and Fig. S3). Sites were established across elevational gradients on all three mountains, and thoroughly surveyed in order to provide genuine presence-absence data, meaning we were certain that undetected species were not present at the site. Standardized survey methodology at each site consisted of 6 h of "timed searches," in which we walked in a zigzag pattern through the understory, stayed within 50 m.a.s.l. of the target elevation, and for each of twelve separate 30-min windows, recorded all species encountered and the total number of individuals per species (of *Elaphoglossum* and *Peperomia*). Species-accumulation curves (SACs) were created using EstimateS software[61–63], using each 30-min window as a sample, and actual diversity was estimated from the asymptote of the curve in order to confirm that the number of species detected at each site was approximately equal to the estimate of true total diversity for the site. Fig. S11 shows, as an example, the SACs for all sites on Baru. In Monteverde, much more extensive occurrence surveys were carried out from 2013 to 2016, using alternative methods including tree climbing, ground plots, and targeted surveys of fallen trees. These alternative methods were used to confirm that timed walks were the most effective method of detecting the species present at a site. In all cases, 6-h timed walks found the greatest number of species and took the least amount of time among the alternative methods. The total number of species detected from 6-h timed walks plus additional incidental records from the site, i.e., species observed outside of the timed walks (usually 1–3 additional species) equaled the estimated diversity calculated by EstimateS for nearly all sites, providing confidence that our species list for each site was comprehensive for our study groups. Thus, the timed walk method was chosen to most effectively survey sites on the other two mountains, enabling surveys over much larger elevational gradients in a feasible time period. In addition to the timed walks, "incidental records" were collected whenever species of interest were encountered on these mountains outside the survey sites, thus contributing to elevational range data. All

mountains were surveyed intensively for a period of 1–3 years, with multiple trips up and down the elevational transects, collecting incidental records on many trails other than just those containing the survey sites. Latitude, longitude and elevation of all sites and incidental records were recorded with a Garmin GPS with accuracy <20 m.

**Elevational range surveys statistics and analysis.** To confirm that surveys accurately captured the full diversity at all sites, species-accumulation curves were created and analyzed for each survey site. Analytical methods are described above.

To calculate species' range sizes along each elevational transect, species ranges were both interpolated and extrapolated as in Colwell et al. (2008) (3). Extrapolation means that a species' range limit was assigned as the midpoint between its last recorded presence and its first recorded absence. Interpolation means that if a species' range extent spanned an elevation along a given transect, e.g., it was detected above and below, but it was not detected at that elevation, it was considered present at that elevation.

To calculate extinction risk, we used species' elevational ranges as measured in the field as a proxy for their climatic tolerances, since the transplant experiment suggested that these species cannot survive much outside their native range, and especially at lower elevations. Ranges were used to extrapolate climatic tolerances in two alternate ways: based on a species' aggregate range across all three mountains, or based on each population's elevational range on a single mountain. In the aggregate scenario, each population of a given species across the three mountains was considered to have the same climatic tolerance, spanning the breadth of elevations at which we recorded it in the field across all populations. Thus, the species' aggregate ranges yield a larger estimate of their climatic tolerance. In the individual population scenario, the populations on each mountain are considered to be restricted to the elevations where they occur on that mountain.

We used the elevational lapse rate we measured at Monteverde (5.33 °C per 1000 m elevation) to estimate how much of an elevational shift would be needed to maintain a specific temperature profile with a given amount of climate warming. We used these elevational shifts to estimate how much the lower-elevational boundary of a species range would shift upward. Under a "no dispersal" scenario, we assumed the upper boundary of species distributions would remain static, leading to range contractions. Under a "full dispersal" scenario, we assumed upper boundaries would shift upward by an equal amount as lower boundaries, unless upper boundaries were limited by the total elevational extent of a mountain.

Two alternate extinction measures are calculated: Population extirpations and species extinctions. In the former, we measure the number of individual populations expected to lose all their range on the mountain where they occur. For species extinctions, a species is only considered extinct (in this region) if all of its populations in all the mountains where it occurs lose all of their range.

Finally, we calculated the average breadth of the elevational range remaining for those species and populations that are not projected to be extirpated. Even if populations remain extant, range loss would put them at higher risk of extinction due to stochasticity, so a decline in average range size indicates greater vulnerability of the assemblage to future loss. Extant species and populations will lose elevational range either due to range contraction from the lower boundary without corresponding expansion (in the no dispersal scenario), or when they abut the tops of their individual mountains (in the dispersal scenario).

**Ethics and inclusion statement**
This study did not involve local academic researchers from Costa Rica or Panama. It did involve help in the field, advice, and management from significant numbers of individuals associated with the public and private reserves in which the fieldwork was carried out, as well as private individuals who worked with the authors as field assistants, and who are listed in Acknowledgements. The local relevance of the research was discussed extensively with local conservation organizations, especially the Monteverde Cloud Forest Reserve and the Monteverde Conservation League. The research was approved by the relevant branches of the Costa Rican and Panamanian national governments, e.g., their environmental ministries. All collections and materials remained in the host countries. We have cited all relevant local research of which we are aware.

**Reporting summary**
Further information on research design is available in the Nature Portfolio Reporting Summary linked to this article.

## Data availability
All data for this study were collected in the field by the authors. The transplant data generated in this study have been deposited in an Open Science Framework repository and can be accessed at: https://osf.io/f54zy/?view_only=18f10d58704c4d9b8d3f13c0cc3cc6f1. The range survey data generated in this study are provided in the Supplementary Information (Tables S6–8). Source data are provided with this paper.

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

## Acknowledgements
We are indebted to many individuals and organizations who aided and enriched this research in a variety of ways, including research permits and permissions. We received research permits from Sistema Nacional de Areas de Conservación (SINAC) in Costa Rica from 2013–17, and the Ministerio de Ambiente in Panama in 2016. Data were collected on properties belonging to the following individuals and organizations: The Monteverde Cloud Forest Biological Preserve (Tropical Science Center), Reserva Curi-Cancha, Bosque Eterno de los Niños, the Leitón family, Frank Joyce & Katy van Dusen, Braulio Carrillo National Park (Costa Rica), La Selva Biological Station (Organization for Tropical Studies), La Fortuna Reserve (Smithsonian Tropical Research Institute), Volcán Barú National Park (Panama). We were aided in logistics by the Monteverde Institute, the Monteverde Cloud Forest Biological Preserve, the Organization for Tropical Studies, and the Smithsonian Tropical Research Institute. We are grateful for field assistance from Phoebe Hopkins, Cecilia Cerrilla, Yesenia Valverde, Emmaline Suchland, Adam Moreno, Krystal House, Tony Obando, Eladio Cruz, Juan Huertas, David Miranda, Chalo Miranda, Bady Garcia, Fernando Obando, Chico Fuentes, Mino Fuentes, Elier Zuñiga, Jocksan Mata, Marcos Molina, and Gilbert Hurtado. We received generous help learning the taxonomy of our study species from Robbin Moran, Fernando Bittencourt Matos, Alexander Rojas, and Guido Mathieu. The manuscript benefited from helpful discussion over the course of the study with Jon Witman, Erika Edwards, and Sohini Ramachandran. Funding for this research was provided by the following institutions and grants: National Science Foundation (NSF) Graduate Research Fellowship 1644760 (ECH). NSF IGERT grant DGE 0966060 (ECH, PI: David Rand). Institute at Brown for the Environment and Society (IBES), Graduate Research and Travel grant (ECH). IBES Graduate Fellowship (ECH). Brown University Department of Ecology & Evolutionary Biology, Doctoral Dissertation Improvement Grant (ECH).

## Author contributions
Conceptualization, methodology, visualization, funding acquisition, project administration, writing—original draft, and writing—review and edition: ECH and DFS, Formal analysis and investigation: ECH, and Supervision: DFS.

## Competing interests
The authors declare no competing interests.
