## [Peer Review File · Nature Communications]

Experimental evidence of climate change extinction risk in Neotropical montane epiphytesREVIEWER COMMENTS

Reviewer #1 (Remarks to the Author):

This is an exciting and elegant experimental study that addresses important questions about the response of wet forest herbaceous taxa to climate change across elevational gradients. The authors experimentally tested the climate tolerances of 15 species across two genera of herbaceous taxa along a 500m gradient on one mountain, and modeled extinction risk for taxa across elevations of three mountains. Per the authors, this is the first experimental study of extinction risk from climate change for tropical montane species. Of note is that this study validates existing predictions and niche models on extinction risk of tropical montane species and will appeal to researchers across fields.

This study has the potential and promise to strengthen our understanding of the movement of plant species, and highlights the urgency with which the world needs to act to mitigate climate change. This is exactly the kind of in situ experimental research needed to understand how species will respond to climate change. While more experimental validation is needed, these data allow researchers to begin testing more complex questions such as how communities will respond to climate change and how biotic interactions may be disrupted, etc.

As presented, the manuscript needs a stronger theoretical framework to ground the paper. As the authors note, this is the first experimental study to validate tropical montane extinction models and the need for these experiments needs to be highlighted and a stronger argument for its study made. The authors should frame the paper more strongly, perhaps on narrow niches, range shifts, climate change driven range shifts, herbaceous taxa, etc. Further, a more robust discussion on how species may respond to climate change along elevation gradients will help the argument.

Overall, this manuscript feels like two papers, one on a survey across three elevational gradients and one on a transplant study. It is easy to get lost in the supplemental data and I encourage the authors to rethink organization.

Does the work support the conclusions and claims, or is additional evidence needed?

There is not enough information to assess whether the findings are sufficient because the theoretical framing of the paper is lacking. The paper needs to be anchored. Further, a deeper presentation of their system is needed. There is a deep literature on tropical montane forests and predicted impacts of climate change on tropical montane systems, as well as the importance of and problems with transplant studies, but this discussion is missing.

Are there any flaws in the data analysis, interpretation and conclusions? Do these prohibit publication or require revision?

The authors need to be cautious of the application of their findings across all tropical plants, as indicated in the title. The experimental study of climate range limits of two herbaceous, epiphytic genera must be carefully applied. While there was tremendous work involved with the study, it has limitations in scope.

While the supplemental data are rich, it is often unclear what results are presented for which study.

Is the methodology sound? Does the work meet the expected standards in your field?

There is tension between what seems the aim of the paper, measuring extinction risk across 15 species in two genera of tropical herbaceous taxa, and the large surveys. The transplant study is the crux of the paper and those findings are used to model extinction risk across all three elevation gradients; however as presented, this is confusing. The link between the transplant study and the survey needs to be clearer and the survey emphasis diminished.

In addition, the reasoning for transplant experiments within a tight elevation band (500m, 1100-1600m) only, needs to be explained. The results tell us how species between 1100-1600m will respond to changes in climate, and how this can be applied across taxa at 30m needs unpacking. Further, why *Peperomia* and *Elaphoglossum*? Why non canopy specialist

epiphytes? Why Monteverde?

Is there enough detail provided in the methods for the work to be reproduced?

The methods within the manuscript are vague whereas those in the supplementary file are more robust with only a few areas that need clarification. However, there is often confusion as to what is the transplant study and what is the survey. Some figure legends need clarification.

Editorial comments: I have editorial comments and questions on the manuscript itself.

Reviewer #2 (Remarks to the Author):

The work by Hollenbeck and Sax details an ambitious experimental study investigating epiphyte survival under various climate change scenarios along Central American mountain ranges. They find that as epiphytes are transplanted outside of their home elevation, survival declines and they attribute this to changes in climatic conditions. The fieldwork component of this study sounds daunting, and I applaud their efforts on working in a severely understudied taxonomic group under likely difficult field conditions while still achieving reasonable sample sizes and replications. The data generated on such a poorly studied group is valuable in its own right.

They pair this experimental work with survey data from additional mountains in Costa Rica and Panama. These surveys are used to map current distributional ranges and make predictions about future range contractions based on simple climate alteration scenarios along with species dispersal status (can or cannot disperse).

While I don't doubt that epiphyte survival outside of the native location is driven by environmental conditions, I do find some issue with the presentation of the findings of the paper. A good portion of their findings hinges on the experimental component. The point that concerns me most is the lack of attention given to moisture type across the transplant sites; falling precipitation or fog-capture. Their approach is to measure leaf wetness days as

a measure of moisture. This seems reasonable at first glance but when we think about moisture in terms of falling precipitation versus fog capture, this measurement of moisture can become problematic. Indeed, based on the figures (e.g., Fig 2) cloud immersion appears to be more prevalent and reaches lower elevations on the Atlantic side than the Pacific side. Moving individuals into the higher fog-capture environment of the Atlantic side could cause higher mortality due to plants inability to conduct gas exchange or reduced PAR when covered in water, not necessarily due to the amount of moisture available.

This issue with the type of moisture does not invalidate the findings as they found lower survival in wetter sites regardless of precipitation type. Rather I would like to see some attention given to how cloud base would play in their predictions of future range contractions or extinctions. Cloud base is hard to predict but some information is available to help with this topic (Bruijnzeel et al 2013 – Tropical montane cloud forests: science for conservation and management). Furthermore, there findings aren't completely novel, as stated in the paper. Although they are difficult to find and not well cited, Josh Rapp and Miles Silman at Wake Forest conducted a similar experimental study. These studies should probably be included in this paper

<https://www.ncbi.nlm.nih.gov/pmc/articles/PMC4133766/>

<https://www.proquest.com/docview/851885915?pq-origsite=gscholar&fromopenview=true>

In addition to the above concerns I provide more detailed concerns below.

Suggest changing the title to 'tropical epiphytes' instead of the broader 'tropical plants' category as the study only address epiphytes

L16-17 Can this sentence be reworded to get at what actually causes them to die off outside of their elevation band? It isn't necessarily elevation and associated atmospheric concentration that limits them, but rather changes in temperature and humidity associated with elevation. So temperature/humidity are the direct cause and vary with elevation.

L34 comma after dispersal

L48-50 Make clear which part of the study was across the three mountains (surveys) and which was specific to Monteverde (transplants)

L93-95 – Report the % of populations that fall under this category like done in L99 for small ranged populations

Fig 1 – black text on maps is difficult to read in left panel. Need to make clear that two transects were studied at Monteverde? Although looking through the methods, it is unclear where four transects occur. I see just a single transect on each mountain. Figure S1 seems to confirm this.

Figure 2 panel B – If a Large X signals all species had zero survival, then how can there also be small x in the same grid cell. Particularly the two leftmost grid cells and the right most

Fig 3 – unclear what the subbox found in the top left of panels A-D represent? Is this the mountaintop?

L411-415 – give some summary stats in text here on how many species had less than 20 individuals per directions, including the absolute minimum.

Reviewer #3 (Remarks to the Author):

The manuscript by Hollenbeck & Sax presents evidence for the impact of potential climate change on extirpation and extinction of 67 species in two genera of tropical epiphytes in montane forests in Costa Rica and Panama. Through distributional surveys and experimental transplant manipulations over three years the authors clearly show that changes in temperature and aridity (both future effects of climate change) across elevational gradients may have substantial effects on species survival under a number of climate change scenarios. In their investigations the authors have provided another example in a long line of evidence, which they carefully cite, for the potential influence on tropical vegetation of temperature increase due to rising CO₂ levels in the atmosphere. This investigation is an admirable initial effort that could be greatly expanded in the future to produce and

interpret significant results.

As currently written, the investigation is much too limited in scope and interpretation to be published in present form. To extrapolate the results of such a restricted number of species and genera of a single life form (epiphytes) to the tropical montane flora, which includes terrestrial herbs, shrubs, lianas, and trees, is not warranted at this point in time. The authors justify their selection of epiphytes as the sole focus of investigation on the basis that these plants are experimentally "amenable to transplant." Unfortunately this rationale alone is not sufficient to expand their results to "tropical plants" in general. The methodology of conducting surveys and transplant experiments is reasonable. However, until these approaches are applied to a broader sample of life forms and taxonomic diversity of tropical plants, the results as given should only be considered as "preliminary" at most.

Their main conclusions that "in general, movements to drier and/or warmer climates caused greater and more consistent mortality than did cooler and/or wetter climates" and "reduced survival at drier sites appears to be caused by abiotic stress, and at wetter sites by an interaction between abiotic and biotic stress" would not be a surprise to any gardener who has ever transplanted a tropical species from one site to another. Many other statements in the manuscript are equally obvious to someone who has cultivated plants in a greenhouse or outdoor garden.

In addition to the significant expansion of sampling, several other comments the authors should consider in preparing a future manuscript include:

- 1) Expanding citations to encompass more recent publications on the impact of climate change on tropical biota. Of their current 35 references, only 4 have been published in the last five years, with most 10-20 years old.

- 2) Figures 2-4 are informative but each contains a giant amount of information that may best be separated into several illustrations that are more accessible for the reader.

- 3) The authors make a good point that the slow growth of their epiphytic study species may

preclude local adaptation as a successful response to climate change, but they should note that a number of recent papers have demonstrated that adaptations in agricultural weeds to rising temperatures and drier climates may occur at an unexpectedly rapid rate.

This paper will deserve publication in the future when a broader set of tropical taxa and life forms are surveyed. Perhaps a more specialized journal would also be more appropriate.

Summary: We thank the reviewers for their detailed and thoughtful comments. We have taken these comments to heart and now provide what we believe to be a more compelling manuscript. Our biggest changes are two-fold: first, providing a richer theoretical underpinning and empirical justification for our study design and location, and second, better framing and justifying why the combination of the two different elements of this study (the reciprocal transplant experiment and the surveys of species distributions across multiple tropical mountains) is so important in providing insight into the risks of extinction these species face from climate change. We hope the reviewers agree that the new manuscript, incorporating many of their ideas and comments, is much improved from the original. Below we provide a point-by-point response to the various thoughts and feedback provided by you and the reviewers.

REVIEWER COMMENTS

Reviewer #1 (Remarks to the Author):

This is an exciting and elegant experimental study that addresses important questions about the response of wet forest herbaceous taxa to climate change across elevational gradients. The authors experimentally tested the climate tolerances of 15 species across two genera of herbaceous taxa along a 500m gradient on one mountain, and modeled extinction risk for taxa across elevations of three mountains. Per the authors, this is the first experimental study of extinction risk from climate change for tropical montane species. Of note is that this study validates existing predictions and niche models on extinction risk of tropical montane species and will appeal to researchers across fields.

Thank you for this positive response to our work.

This study has the potential and promise to strengthen our understanding of the movement of plant species, and highlights the urgency with which the world needs to act to mitigate climate change. This is exactly the kind of in situ experimental research needed to understand how species will respond to climate change. While more experimental validation is needed, these data allow researchers to begin testing more complex questions such as how communities will respond to climate change and how biotic interactions may be disrupted, etc.

Thank you again.

As presented, the manuscript needs a stronger theoretical framework to ground the paper. As the authors note, this is the first experimental study to validate tropical montane extinction models and the need for these experiments needs to be highlighted and a stronger argument for its study made. The authors should frame the paper more strongly, perhaps on narrow niches, range shifts, climate change driven range shifts, herbaceous taxa, etc. Further, a more robust discussion on how species may respond to climate change along elevation gradients will help the argument.

These comments were very helpful. I think we fell into the classic trap of being too close to our own work and as a consequence did not previously do a good enough job of providing a theoretical and empirical underpinning for the work. We have completely revised and significantly expanded the introduction of the manuscript. In this effort we have expanded our coverage of uphill movements of species as a climate adaptation strategy. We have better framed the paper around the paucity of experimental evidence to support the contention that tropical montane species have narrow climate tolerances – a contention that follows directly from the common application of species distribution models, but which has previously been in need of more direct experimental support. We also frame the paper more strongly around the paucity of detailed elevational surveys of species distributions in tropical mountains, and explain why

estimating extinction risk is so difficult without detailed distributional data. We also specifically highlight manuscripts that have explicitly called out a need for the sort of work we have conducted.

Overall, this manuscript feels like two papers, one on a survey across three elevational gradients and one on a transplant study. It is easy to get lost in the supplemental data and I encourage the authors to rethink organization.

This is perhaps the single most helpful comment we have received. In reconsidering our manuscript, it became clear to us that we had not explained why these two aspects of the work (the experiment and the surveys) dovetail so critically together. To address this, we made a number of changes to the manuscript. First, as described in the comment above, we expanded the introduction, specifically calling out these two issues and the call for both types of work that have been issued by other researchers. As we now better explain in the main text of the manuscript, it is nearly impossible to develop well-supported estimates of extinction from climate change, ones that depend critically on where species occur on individual mountains, if the occurrence of those species isn't clearly known; we also explain that such detailed records are rarely available in tropical montane settings, which is why other researchers have called for detailed work of this nature. Further, we explain that estimates of species loss are better informed by being able to compare local extinction/extirpation risks (on a single mountain) with broader risks of species loss across multiple mountains – something that is only possible for tropical montane species after detailed surveys are conducted across multiple mountains.

Second, we revised Figure 1, so that both the experiment and the surveys are apparent to the reader right away, and visually – a change that we think helps to better identify these two portions of the work as being integral to the overall study. We also tried to better streamline and integrate the main text and supplementary material. To that end, we have better incorporated some of the main results from the supplementary material into the main text itself. We also removed some details from Figure 4, around loss of total elevational range for species not at immediate risk of extinction from climate change, and moved that to the supplementary material – in order to keep the reader focused on our most important points in the main text. In the supplementary material itself, we removed a figure (about elevational patterns of diversity) that while interesting, is not critical to the focus of the present manuscript.

Does the work support the conclusions and claims, or is additional evidence needed?

There is not enough information to assess whether the findings are sufficient because the theoretical framing of the paper is lacking. The paper needs to be anchored. Further, a deeper presentation of their system is needed. There is a deep literature on tropical montane forests and predicted impacts of climate change on tropical montane systems, as well as the importance of and problems with transplant studies, but this discussion is missing.

We agree. We have better anchored the theoretical framing for the paper (as described in the comments above). We have also expanded our discussion of the literature on tropical mountains, better explaining what sorts of work has previously been conducted: SDM-work, SDM-work informed by surveys, experiments, etc. We have also added some text and citations around the pitfalls that can be associated with transplant studies, including a discussion of why our specific work addresses many of the most common experimental design limitations.

Are there any flaws in the data analysis, interpretation and conclusions? Do these prohibit publication or require revision?

The authors need to be cautious of the application of their findings across all tropical plants, as indicated in the title. The experimental study of climate range limits of two herbaceous, epiphytic genera must be

carefully applied. While there was tremendous work involved with the study, it has limitations in scope.

Yes, you are correct. We have toned down the language around the generality of the work. We have also added in more references and text (to the final paragraphs of the main text) better contextualizing where our findings sit relative to other work conducted to date, which helps to inform the issue of the generality of our findings.

While the supplemental data are rich, it is often unclear what results are presented for which study.

We have addressed this concern in a couple ways. First, we have slightly expanded our coverage of results in the main text. We think this helps to make more clear how the principal findings of the work (reported in the figures in the main text) are supported by the additional findings we report in the supplementary text. Second, we tried to streamline the supplementary material, deleting one figure entirely, combining two figures, and making other small changes that aim to increase the clarity and supporting nature of the supplementary material.

Is the methodology sound? Does the work meet the expected standards in your field?

There is tension between what seems the aim of the paper, measuring extinction risk across 15 species in two genera of tropical herbaceous taxa, and the large surveys. The transplant study is the crux of the paper and those findings are used to model extinction risk across all three elevation gradients; however as presented, this is confusing. The link between the transplant study and the survey needs to be clearer and the survey emphasis diminished.

We have addressed this concern in our comments above, but in brief we have better explained and highlighted why these two aspects of the work dovetail together in critical ways, enabling a consideration of extinction risk across multiple mountains. We agree in part that the survey emphasis should be diminished, but only with respect to findings from the survey that are not relevant to issues of extinction; in this vein, we removed a supplementary figure that described elevational patterns of species richness across the three surveys; we have retained all aspects of the survey work that are utilized to leverage the experimental data to inform extinction risk.

*In addition, the reasoning for transplant experiments within a tight elevation band (500m, 1100-1600m) only, needs to be explained. The results tell us how species between 1100-1600m will respond to changes in climate, and how this can be applied across taxa at 30m needs unpacking. Further, why *Peperomia* and *Elaphoglossum*? Why non canopy specialist epiphytes? Why Monteverde?*

These are all good questions. Some of this was described previously in our detailed methods section, but not well enough. Consequently, we have added material that addresses most of these points in the main text. We now clearly explain why the experimental work was done at Monte Verde, which is for a host of reasons, but most critically because the trails and reserve area are situated in a way that enabled us to explore species distributions on both side of the continental divide (where quite different environmental gradients exist, which was useful for our survey work), because we could examine meso-scale site differences in aridity at a relatively constant set of elevations/temperatures, which paired with a classic elevational transect, allowed us to have an experimental design with two distinct environmental gradients, i.e., one that controlled for temperature and just varied in aridity and another in which temperature and moisture covaried. This type of experimental setup was important in allowing us to examine the relative influence of changes in aridity. Further, we have now better highlighted in our results of the main text why 500m between the highest and lowest transplant sites provided a truly stark contrast in conditions – with such pronounced increases in mortality for species transplanted from the highest to the lowest sites. This result speaks to just how devastating a projected 600m shift in environmental conditions

(corresponding to a moderately-high warming scenario) is likely to be for species in this system. Finally, we better explain why we selected these two genera for study, which are diverse, have characteristics that allow a trained naturalist to identify many of these species in the field (unlike some other epiphyte groups), and which provide two groups from very different plant lineages (i.e., ferns vs. flowering plants).

Is there enough detail provided in the methods for the work to be reproduced?

The methods within the manuscript are vague whereas those in the supplementary file are more robust with only a few areas that need clarification. However, there is often confusion as to what is the transplant study and what is the survey. Some figure legends need clarification.

We have worked to improve the figures (most notably changes to Figure 1 and some changes to supplementary figures) and add more detail to the main text, which together help to address this concern.

Reviewer #1 (Remarks to the Author from an annotated copy of the manuscript):

I think it may be more effective to present what is done and then why it is problematic, perhaps: Niche models and data repositories of species occurrences have been used to predict potential impacts from climate change, however they are not robust to potential mismatches between the scale of available climate data, the spatial accuracy of available distribution data, and the high, abrupt heterogeneity in environmental conditions that occur in tropical mountains. Direct surveys coupled with experimental transplantation avoid these issues and directly measures/quantitatively assess survival both within and beyond native distributions and In tropical wet forest, epiphytes are an ideal model for such assessment as they are disconnected from the forest floor, perching on branches and trunk, and easily moveable, fast growing, and...

Based on the comments provided in your review we have addressed these specific comments from the annotated manuscript notes. In summary, we now provide (in the revised manuscript) a much stronger justification for the central premise of the work and we now provide a much stronger justification for the study site and study organisms.

More explanation for why they [epiphytes] are a good model group is needed. Most do not know these the diversity of taxa, their small size, herbaceous tendency, etc.

Yes, indeed, this is correct. Our revised manuscript provides a much better explanation for why epiphytes are a useful model group for study.

from one site, multiple sites. Is this common garden? This [experimental] setup is vague and more is needed to follow on.

We have now bolstered our experimental description in the manuscript and (based on your other comments) worked to improve the value the figures provide.

vague. What does this ["struggled to survive"] mean. Lower survivorship, give numbers

The paragraph in question has now been greatly revised. More specificity is now provided and this specific question has been addressed.

what does this mean? [i.e., showing an effect of population of origin]

While we haven't changed the overall sentence in question, it is now embedded in a more focused paragraph that we believe helps to better clarify our meaning, which in this case is that for the widespread species in question, not all of its populations performed equally when transplanted.

How did you deduce this? [Reduced survival at drier sites appears to be caused by abiotic stress, and at wetter sites by an interaction between abiotic and biotic stress] Be careful, this paragraphs feels like a list and should be synthetically argued

Thank you. To address this we have now added more detail to this paragraph, included references to the literature to better contextualize our results relative to other studies, and included additional potential explanations for observed changes.

This concept of local adaptation needs to be developed.

We now better contextualize how a wide variety of alternative factors could increase or, alternatively, decrease the risk estimates we have provided. We now list, and better contextualize, local adaptation among these other factors.

what is this? Where are the data that support slow growth? What are you comparing this rate too? Compared to a tree they [i.e., epiphytes] are fast. Do we see these data?

We have now included a reference about epiphyte's "slow" growth rates, one that is consistent with the qualitative patterns we observed. Since this sentence is addressing a fairly ancillary point, we think this reference is probably sufficient. We debated including the data on growth rates (which we do have), but the supplementary material is already so large that we felt on balance like it wasn't worth including. If you feel strongly about this then we could add the growth rate data to the supplementary material.

It would be helpful [regarding Fig. 1] to have the range laid out early.

We have added this information to the figure.

Unclear why these sites [referring to sites selected for the transplant experiment and shown in Figure 2]. How far apart in meters. What is their elevation? What does encompassing mean here? Do you mean "that varied in temperature, precipitation, and elevation." This is vague. Also, establishing early in the ms that elevational changes are temperature dependent makes it possible to refer only to temperature.

The questions you've asked here are in reference to a panel that was moved in the revised manuscript and is now within Figure 1. We have reworked the writing of the sentences in question to make the wording more specific. We have also added more detail to the main text about these differences in the transects and why these specific sites were selected for the experiment – and better explained why these two distinct transects provide some unique leverage for understanding these species.

only peperomia and elaphoglossum? [for "individuals per species"]

Yes, that is correct. We have added a parenthetical statement to the text to make this clear.

What is "true diversity"? General diversity stats somewhere would be helpful-# of elaphoglossum total per site, etc. the species list is helpful, but the totals are needed.

We have replaced the phrase "true diversity of" with "species present at". We considered keeping a table that lists site diversity explicitly, but on balance given the size of the supplementary material, we have not

included such a table in the revised supplementary material. We have, however, listed the distributions of our study species across study sites, so the interested reader can easily tally these numbers if they are interested. We think this is sufficient. However, if you or the editor feels strongly about this then we could add another supplementary table that summarizes these data.

It is difficult in the supplementary material and throughout to keep track of which sites are where. For each of the three elevational gradients there are sites 1-6. Perhaps a solution is not to number them here...[in what had been Fig. S1 and now is Fig. S3]

We disagree with this suggestion. We think it is important to keep the site numbers in this figure, particularly as they help to clarify the data presented in Table S7.

You also provided several specific suggestions for text edits.

Thank you for the time you put into this effort. We have extensively rewritten portions of the manuscript, so some of the suggestions were no longer relevant. Other suggestions we accepted, although not necessarily in the exact spot suggested. For instance, we now clarify that *Elaphoglossum* is a genus of fern (as you suggested). In some cases, however, we have opted to stick with particular word choices that we think best reflect the tone and meaning we are trying to set; for instance, we are continuing to use the word “conjectured” because we think it best reflects the meaning we are trying to communicate in that particular sentence. We have not provided a word-by-word response to specific wording suggestions, but we can if the editor feels that is important.

Reviewer #1 (Remarks to the Author from an annotated copy of the reporting summary):

You had six points/clarifications listed in the annotation to the reporting summary. These included:

- *Need information on wooden scaffold*
- *More specificity on the number of individuals transplanted*
- *Were scaffold were placed*
- *Were these canopy or understory species*
- *Being sure to be consistent in number of individuals moved per species*
- *Questions about epiphytes were surveyed (e.g., were binoculars used)*
- *Clarifying the intent of the six-week period after attaching epiphytes to scaffolds, but before moving the scaffolds*

We believe that these comments are addressed in the text of the revised manuscript. All of the reviewers’ questions are answered in detail in the Methods section of the manuscript. So, for example, we describe the wooden scaffolds, where they were placed in the experiment, what sort of epiphyte species were surveyed, etc. We have tried our best to simplify the text in the reporting summary and removed specific mentions of the number of individuals transplanted - as this is covered in depth in the revised manuscript.

Reviewer #2 (Remarks to the Author):

The work by Hollenbeck and Sax details an ambitious experimental study investigating epiphyte survival under various climate change scenarios along Central American mountain ranges. They find that as

epiphytes are transplanted outside of their home elevation, survival declines and they attribute this to changes in climatic conditions. The fieldwork component of this study sounds daunting, and I applaud their efforts on working in a severely understudied taxonomic group under likely difficult field conditions while still achieving reasonable sample sizes and replications. The data generated on such a poorly studied group is valuable in its own right.

Thank you for these positive comments and for recognizing the immense amount of field work involved in generating these data.

They pair this experimental work with survey data from additional mountains in Costa Rica and Panama. These surveys are used to map current distributional ranges and make predictions about future range contractions based on simple climate alteration scenarios along with species dispersal status (can or cannot disperse).

While I don't doubt that epiphyte survival outside of the native location is driven by environmental conditions, I do find some issue with the presentation of the findings of the paper. A good portion of their findings hinges on the experimental component. The point that concerns me most is the lack of attention given to moisture type across the transplant sites; falling precipitation or fog-capture. Their approach is to measure leaf wetness days as a measure of moisture. This seems reasonable at first glance but when we think about moisture in terms of falling precipitation versus fog capture, this measurement of moisture can become problematic. Indeed, based on the figures (e.g., Fig 2) cloud immersion appears to be more prevalent and reaches lower elevations on the Atlantic side than the Pacific side. Moving individuals into the higher fog-capture environment of the Atlantic side could cause higher mortality due to plants inability to conduct gas exchange or reduced PAR when covered in water, not necessarily due to the amount of moisture available.

This is an interesting point. Perhaps mortality of species moved to higher elevations, where conditions are wetter (and cloudier) is driven by issues like gas exchange. Species moved to wetter environments were more likely to rot (an interaction between abiotic conditions and biotic stress), but if gas exchange was also impaired then it might have weakened these high-elevation transplants in a way that put them at greater risk. We have added this possibility to our main text. We have also expanded our coverage of leaf wetness, and why a lack of leaf wetness for species moved to the drier sites is likely to be so devastating for some of these species adapted to wetter conditions, which can depend on foliar water uptake (which we now support with citations) to maintain their water balance.

This issue with the type of moisture does not invalidate the findings as they found lower survival in wetter sites regardless of precipitation type. Rather I would like to see some attention given to how cloud base would play in their predictions of future range contractions or extinctions. Cloud base is hard to predict but some information is available to help with this topic (Bruijnzeel et al 2013 – Tropical montane cloud forests: science for conservation and management). Furthermore, their findings aren't completely novel, as stated in the paper. Although they are difficult to find and not well cited, Josh Rapp and Miles Silman at Wake Forest conducted a similar experimental study. These studies should probably be included in this paper

<https://www.ncbi.nlm.nih.gov/pmc/articles/PMC4133766/>

<https://www.proquest.com/docview/851885915?pq-origsite=gscholar&fromopenview=true>

Thank you for these comments. In hindsight, we feel badly that the Rapp and Miles experimental study wasn't cited. We have now cited it and we have cited several other experimental studies and their findings. While all of these earlier works are important, and add to our collective knowledge, they each had some limitations with respect to length of study, number of replicates, number of study species, etc.

We now use this earlier body of work to help better contextualize our own work and better explain the importance of the work we conducted.

We did consider trying to better account for how changes in cloud cover might play out on these mountains. Our results suggest, given the sensitivity to changes in aridity that we showed in our experiments, that if dramatic, non-linear changes in cloud cover resulted over and beyond the simple elevational shifts we model in the paper, that extinction rates would likely be even higher. We have now added this point to our discussion of extinction risks. We have also added a recent citation (Helmer et al. 2019) that models these risks in the Neotropics. We do not feel, however, that we can specifically quantify this risk in our manuscript given the complexity of this issue. We think future work that specifically delved into this issue would be useful, but that such work would likely be a stand-alone manuscript given the complexity of the topic.

In addition to the above concerns I provide more detailed concerns below.

Suggest changing the title to ‘tropical epiphytes’ instead of the broader ‘tropical plants’ category as the study only address epiphytes

We have changed the title in such a way that this previous comment is no longer a concern.

L16-17 Can this sentence be reworded to get at what actually causes them to die off outside of their elevation band? It isn’t necessarily elevation and associated atmospheric concentration that limits them, but rather changes in temperature and humidity associated with elevation. So temperature/humidity are the direct cause and vary with elevation.

We considered your suggestion here. We do attempt to get at this mechanism in the main text, but in the summary paragraph we believe that our simple description of the empirical pattern we observed is best for the paragraph’s clarity.

L34 comma after dispersal

We modified the first paragraph of the main text enough that this specific suggestion is no longer relevant.

L48-50 Make clear which part of the study was across the three mountains (surveys) and which was specific to Monteverde (transplants)

We have addressed this in the text and with an updated Figure 1.

L93-95 – Report the % of populations that fall under this category like done in L99 for small ranged populations

This is an excellent suggestion and we have added this information to the main text.

Fig 1 – black text on maps is difficult to read in left panel. Need to make clear that two transects were studied at Monteverde? Although looking through the methods, it is unclear where four transects occur. I see just a single transect on each mountain. Figure S1 seems to confirm this.

We have updated Figure 1, which now makes each of these issues clear.

Figure 2 panel B – If a Large X signals all species had zero survival, then how can there also be small x

in the same grid cell. Particularly the two leftmost grid cells and the right most

Thank you for pointing this out. We were trying to make a distinction between the number of species in the grid cell and the fate of the species in that grid cell, but the way we did it was confusing (and probably not necessary). We have removed the large X's from the figure and believe this improves the figure clarity.

Fig 3 – unclear what the subbox found in the top left of panels A-D represent? Is this the mountaintop?

We have added text to the figure legend to make the intention of the boxes in the top left of each panel more clear. Additionally, we noticed that the original color shading of those boxes, which was referenced in the figure legend, disappeared in the pdf proof that was sent to the reviewers. We apologize if this impacted the clarity of the figure, and we will attempt to confirm that the figure colors are accurately reflected in the resubmission.

L411-415 – give some summary stats in text here on how many species had less than 20 individuals per directions, including the absolute minimum.

We have added some summary statistics, reporting the mean and range in number of individuals per species per direction of transplant used in the experiment; we also report the specific number of individuals for each individual case in the supplementary material.

Reviewer #3 (Remarks to the Author):

The manuscript by Hollenbeck & Sax presents evidence for the impact of potential climate change on extirpation and extinction of 67 species in two genera of tropical epiphytes in montane forests in Costa Rica and Panama. Through distributional surveys and experimental transplant manipulations over three years the authors clearly show that changes in temperature and aridity (both future effects of climate change) across elevational gradients may have substantial effects on species survival under a number of climate change scenarios. In their investigations the authors have provided another example in a long line of evidence, which they carefully cite, for the potential influence on tropical vegetation of temperature increase due to rising CO₂ levels in the atmosphere. This investigation is an admirable initial effort that could be greatly expanded in the future to produce and interpret significant results.

As currently written, the investigation is much too limited in scope and interpretation to be published in present form. To extrapolate the results of such a restricted number of species and genera of a single life form (epiphytes) to the tropical montane flora, which includes terrestrial herbs, shrubs, lianas, and trees, is not warranted at this point in time. The authors justify their selection of epiphytes as the sole focus of investigation on the basis that these plants are experimentally "amenable to transplant." Unfortunately this rationale alone is not sufficient to expand their results to "tropical plants" in general. The methodology of conducting surveys and transplant experiments is reasonable. However, until these approaches are applied to a broader sample of life forms and taxonomic diversity of tropical plants, the results as given should only be considered as "preliminary" at most.

While we do not agree with the conclusion of the paragraph above, we do think that some important points were raised that we have addressed in our revised manuscript. We have added text that better contextualizes the value of our work and which better explains the theoretical and empirical underpinnings for the work we've conducted. Part of this added text explains why epiphytes in particular make for a good study organism and why results that only pertained to them would still be of general

interest (e.g., because they can comprise up to 50% of the vascular plant biodiversity in some cloud forest systems). We agree with the reviewer that we had previously overstated the generality of the work and we have toned down and better contextualized the extent to which our findings may be applicable to other groups.

We do respectfully disagree with the conclusion that the work we conducted is not sufficient in scope to be published in a top tier journal. These sorts of experiments are a tremendous amount of work, which is why by necessity they focus on a limited number of species. Our experimental work addresses many more species, and many more sites, with much greater replication, better controls, and over a much longer time (3 years) than any previous experiment we are aware of in tropical montane systems. Further, while we think the experiment by itself is above any sort of bar for the amount of work conducted, we think when leveraged with detailed species surveys across three different mountains in Central America that what we have produced is truly globally unique and worthy of publication based on its existing data.

Their main conclusions that "in general, movements to drier and/or warmer climates caused greater and more consistent mortality than did cooler and/or wetter climates" and "reduced survival at drier sites appears to be caused by abiotic stress, and at wetter sites by an interaction between abiotic and biotic stress" would not be a surprise to any gardener who has ever transplanted a tropical species from one site to another. Many other statements in the manuscript are equally obvious to someone who has cultivated plants in a greenhouse or outdoor garden.

There is a long line of evidence, generally based on correlative niche models, that suggests that tropical montane species should have narrow climatic niches and tolerances. As we now better explain in the manuscript (in our expanded introduction), these correlative approaches are important, but the accuracy of their conclusions has been in question in the absence of experimental evidence. This had led other researchers to call for such experimental work. We believe our experimental work now helps to bolster the large body of previous correlative work that has been conducted, and also bolsters a handful of smaller-scale experiments that have previously been conducted.

It is not widely appreciated just how often plants are able to grow in climate conditions that differ from those of their native range. For example, there are many tropical plants that are able to grow in conditions that differ from those found in their native ranges (e.g., plants of tropical origin grown in subtropical conditions). More generally, many plants, from many parts of the world, can handle climate conditions that often vary tremendously from those of the native range. An extreme example would be something like Monterey Pine, which in its non-native range has a climatic niche that is roughly 800% larger than its native climate niche (Perret et al.; Naturalized distributions show that climatic disequilibrium is structured by niche size in pines (*Pinus* L.). *Global Ecology and Biogeography* 28: 429-441). It turns out that most plant species examined to date (for which there has been a strong bias towards examining temperate species) are actually able to thrive outside their native range in climate conditions that differ from their native range. Little of this work, however, has been done with tropical species. In cloud forests, in particular, as we suggested above, there is a long line of work that suggests that these species niches might really be as restricted as one would guess solely from their native distributions. Empirically demonstrating this here, as we have done, is important because it contextualizes the very real danger these species face from climate change.

In addition to the significant expansion of sampling, several other comments the authors should consider in preparing a future manuscript include:

1) Expanding citations to encompass more recent publications on the impact of climate change on tropical biota. Of their current 35 references, only 4 have been published in the last five years, with most 10-20

years old.

Yes, you are correct. We have now expanded our coverage of more recent citations.

2) Figures 2-4 are informative but each contains a giant amount of information that may best be separated into several illustrations that are more accessible for the reader.

We have modified and increased the main text in the manuscript, which we think now enables these figures to work more effectively and relate the intended information to the reader. That said, we did somewhat simplify Figure 4, and did some reordering of Figures 1 and 2, all with the aim of improving the clarity of the information provided.

3) The authors make a good point that the slow growth of their epiphytic study species may preclude local adaptation as a successful response to climate change, but they should note that a number of recent papers have demonstrated that adaptations in agricultural weeds to rising temperatures and drier climates may occur at an unexpectedly rapid rate.

This is a good point. We have added some brief text and a citation to acknowledge the possibility of rapid evolution.

This paper will deserve publication in the future when a broader set of tropical taxa and life forms are surveyed. Perhaps a more specialized journal would also be more appropriate.

As we described above, we disagree with these conclusions.

REVIEWERS' COMMENTS

Reviewer #1 (Remarks to the Author):

I appreciate the care the authors took to revise the manuscript and I think many of my initial issues were resolved. This is strong experimental work with findings that will contribute to research on the impacts of climate change on elevational range shifts. While theoretically strong, the paper would be clearer and easier to follow with more editing of the writing. I also strongly recommend that the 3 paragraphs in the discussion on how the assumptions could change outcomes be streamlined.

Comments:

You state early that epiphytes may have narrow ranges but later in the introduction you give ample support that they do. The first is hedging, the second is not.

Be careful as some of the writing is awkward and very passive.

Line 74: Monteverde is an iconic location for studying mesic, tropical montane “cloud forest” ecosystems, and is one of the first places globally where species extinction was ascribed to climate change (35). The mountain’s complex topographical interactions with cloud cover enabled us to examine two distinct climatic gradients, one in which aridity varies but temperature is relatively constant, and another in which aridity and temperature covary (Fig. 1a, Fig. S1). Along these two gradients, we situated six sites where reciprocal transplants and multiple experimental controls were conducted (Fig. S2).

Perhaps:

Monteverde is an iconic location for studying mesic, tropical montane “cloud forest” ecosystems, and is the site of one of the first climate change driven species extinctions (35). The mountain’s complex topographical interactions with cloud cover results in two distinct climatic gradients, one in which aridity varies but temperature is relatively constant, and another in which aridity and temperature covary (Fig. 1a, Fig. S1). Along these two different

gradients we established six reciprocal transplant and multiple control sites (Fig. S2).

I recommended avoiding paragraphs with more than 1 to 2 commas, it makes them cumbersome and diminishes understanding. Note the sentence below has 5 commas—a challenge to anyone!

Line 134: In contrast, mortality at wetter sites was often associated with rotting tissue, a moisture-related biotic pressure involving interactions with microorganisms, which might occur in these species because of a lack of adequate defenses, but could also potentially be accentuated, in species adapted for dry conditions, by stress from reduced gas-exchange ability that could occur when leaves are wet.

Line 207: Sometimes I get lost in the writing: “Because the timing and magnitude of future warming is uncertain, we consider warming along a continuum, while also calling out impacts specifically at two temperatures: 1.5°C, which is achievable only under the most optimistic emission scenarios, and 3.2°C, which is projected for the region under moderately-high emission scenarios, but is likely to be exceeded under highest emission scenarios (44).”

Again, I encourage simplifying some of your sentences from 5 commas and 1 colon. We present extinction risk projections for both 1.5°C and 3.2°C scenarios, low and high emissions.

Line 253: “exceedingly slow growth” As commented earlier, this is vague and not a useful description. Epiphytes are unknown to many, so even though an ancillary point, a number would be helpful. You also cite Spicer and Woods who write: “However, growth rates of epiphytes have not been measured in a wide enough range of taxa from a diverse enough array of environments to definitively conclude this.” They cite Watkins (2006) which is on fern gametophytes.

Figures

The figures in the MS are clearer and more intuitive.

Figure 3D. It would be helpful to explain in the legend “m below mountaintop.” As is, “while D) considers each species’ elevational range as the sum of its ranges on all mountains where it was found.” does not explain the figure well.

Reviewer #3 (Remarks to the Author):

The authors have accomplished an exceptional amount of work in their revision of the manuscript now entitled "Leveraging experiments and elevational surveys across Neotropical mountains to inform extinction risk from climate change." They have also adequately responded to the comments and suggestions of the reviewers. However, many of the original issues with regards to sample size, scope of analyses, and complicated reasoning and presentation remain. A stronger discussion in the conclusion of the limitations of the author's investigation and a broader outline of next steps to expand the scope would make the manuscript appropriate for publication.

REVIEWERS' COMMENTS

Reviewer #1 (Remarks to the Author):

I appreciate the care the authors took to revise the manuscript and I think many of my initial issues were resolved. This is strong experimental work with findings that will contribute to research on the impacts of climate change on elevational range shifts. While theoretically strong, the paper would be clearer and easier to follow with more editing of the writing. I also strongly recommend that the 3 paragraphs in the discussion on how the assumptions could change outcomes be streamlined.

We thank the reviewer for the detailed attention to our writing, and we agree that it is much improved with this round of edits. We have edited all sections of the manuscript to improve the flow of the writing and streamline many parts, including the discussion as mentioned.

Comments:

You state early that epiphytes may have narrow ranges but later in the introduction you give ample support that they do. The first is hedging, the second is not.

We changed the precise language of the first example to say that “[tropical montane species] have narrow ranges.”

Be careful as some of the writing is awkward and very passive.

Line 74: Monteverde is an iconic location for studying mesic, tropical montane “cloud forest” ecosystems, and is one of the first places globally where species extinction was ascribed to climate change (35). The mountain’s complex topographical interactions with cloud cover enabled us to examine two distinct climatic gradients, one in which aridity varies but temperature is relatively constant, and another in which aridity and temperature covary (Fig. 1a, Fig. S1). Along these two gradients, we situated six sites where reciprocal transplants and multiple experimental controls were conducted (Fig. S2).

Perhaps:

Monteverde is an iconic location for studying mesic, tropical montane “cloud forest” ecosystems, and is the site of one of the first climate change driven species extinctions (35). The mountain’s complex topographical interactions with cloud cover results in two distinct climatic gradients, one in which aridity varies but temperature is relatively constant, and another in which aridity and temperature covary (Fig. 1a, Fig. S1). Along these two different gradients we established six reciprocal transplant and multiple control sites (Fig. S2).

This section has been rewritten in accordance with the suggestions.

I recommended avoiding paragraphs with more than 1 to 2 commas, it makes them cumbersome and diminishes understanding. Note the sentence below has 5 commas—a challenge to anyone!

Line 134: In contrast, mortality at wetter sites was often associated with rotting tissue, a moisture-related biotic pressure involving interactions with microorganisms, which might occur in these species because of a lack of adequate defenses, but could also potentially be accentuated, in species adapted for dry conditions, by stress from reduced gas-exchange ability that could occur when leaves are wet.

This sentence has been shortened in accordance with the suggestions.

Line 207: Sometimes I get lost in the writing: “Because the timing and magnitude of future warming is uncertain, we consider warming along a continuum, while also calling out impacts specifically at two temperatures: 1.5°C, which is achievable only under the most optimistic emission scenarios, and 3.2°C, which is projected for the region under moderately-high emission scenarios, but is likely to be exceeded under highest emission scenarios (44).”

Again, I encourage simplifying some of your sentences from 5 commas and 1 colon. We present extinction risk projections for both 1.5°C and 3.2°C scenarios, low and high emissions.

This section has been rewritten in accordance with the suggestions.

Line 253: “exceedingly slow growth” As commented earlier, this is vague and not a useful description. Epiphytes are unknown to many, so even though an ancillary point, a number would be helpful. You also cite Spicer and Woods who write: “However, growth rates of epiphytes have not been measured in a wide enough range of taxa from a diverse enough array of environments to definitively conclude this.” They cite Watkins (2006) which is on fern gametophytes.

We thank the reviewer for pointing out this inconsistency. We have softened our claim of slow growth rates among epiphytes as we cannot find a reliable citation for this contention, although it has been observed in the field.

Figures

The figures in the MS are clearer and more intuitive.

Thank you! The previous round of reviews really helped us improve the figures, and we are grateful for the feedback.

Figure 3D. It would be helpful to explain in the legend “m below mountaintop.” As is, “while D) considers each species’ elevational range as the sum of its ranges on all mountains where it was found.” does not explain the figure well.

Thank you for pointing out this clarification. We have added the following sentence to the figure 3 legend to improve the explanation:

“In panel D, the position on the y-axis measures the distance below the top of the highest mountain where each species was found, rather than total elevation a.s.l., to better illustrate the risk of mountaintop extinction as the three mountains have different peak heights.”

Reviewer #3 (Remarks to the Author):

The authors have accomplished an exceptional amount of work in their revision of the manuscript now entitled "Leveraging experiments and elevational surveys across Neotropical mountains to inform extinction risk from climate change." They have also adequately responded to the comments and suggestions of the reviewers. However, many of the original issues with regards to sample size, scope of analyses, and complicated reasoning and presentation remain. A stronger discussion in the conclusion of the limitations of the author's investigation and a broader outline of next steps to expand the scope would make the manuscript appropriate for publication.

We added a section discussing the limitations of the transplant experiment and its scope, as well as a justification for emphasizing our results in the broader context of knowledge (and current lack thereof) about tropical species and climate tolerance.